# Upstream Regulator Analysis of Wooden Breast Myopathy Proteomics in Commercial Broilers and Comparison to Feed Efficiency Proteomics in Pedigree Male Broilers

**DOI:** 10.3390/foods10010104

**Published:** 2021-01-06

**Authors:** Walter G. Bottje, Kentu R. Lassiter, Vivek A. Kuttappan, Nicholas J. Hudson, Casey M. Owens, Behnam Abasht, Sami Dridi, Byungwhi C. Kong

**Affiliations:** 1Department of Poultry Science & The Center of Excellence for Poultry Science, University of Arkansas, 1260 W. Maple, Fayetteville, AR 72701, USA; klassit@uark.edu (K.R.L.); cmowens@uark.edu (C.M.O.); dridi@uark.edu (S.D.); bkong@uark.edu (B.C.K.); 2Novus International, 20 Research Park, St. Charles, MO 63304, USA; Vivek.Kuttappan@novusint.com; 3School of Agriculture and Food Sciences, The University of Queensland, Gatton, Brisbane, QLD 4072, Australia; n.hudson@uq.edu.au; 4Department of Animal and Food Sciences, University of Delaware, Newark, DE 19716, USA; abasht@Udel.edu

**Keywords:** wooden breast, myopathy, proteomics, feed efficiency, upstream regulator analysis

## Abstract

In an effort to understand the apparent trade-off between the continual push for growth performance and the recent emergence of muscle pathologies, shotgun proteomics was conducted on breast muscle obtained at ~8 weeks from commercial broilers with wooden breast (WB) myopathy and compared with that in pedigree male (PedM) broilers exhibiting high feed efficiency (FE). Comparison of the two proteomic datasets was facilitated using the overlay function of Ingenuity Pathway Analysis (IPA) (Qiagen, CA, USA). We focused on upstream regulator analysis and disease-function analysis that provides predictions of activation or inhibition of molecules based on (a) expression of downstream target molecules, (b) the IPA scientific citation database. Angiopoeitin 2 (ANGPT2) exhibited the highest predicted activation Z-score of all molecules in the WB dataset, suggesting that the proteomic landscape of WB myopathy would promote vascularization. Overlaying the FE proteomics data on the WB ANGPT2 upstream regulator network presented no commonality of protein expression and no prediction of ANGPT2 activation. Peroxisome proliferator coactivator 1 alpha (PGC1α) was predicted to be inhibited, suggesting that mitochondrial biogenesis was suppressed in WB. PGC1α was predicted to be activated in high FE pedigree male broilers. Whereas RICTOR (rapamycin independent companion of mammalian target of rapamycin) was predicted to be inhibited in both WB and FE datasets, the predictions were based on different downstream molecules. Other transcription factors predicted to be activated in WB muscle included epidermal growth factor (EGFR), X box binding protein (XBP1), transforming growth factor beta 1 (TGFB1) and nuclear factor (erythroid-derived 2)-like 2 (NFE2L2). Inhibitions of aryl hydrocarbon receptor (AHR), AHR nuclear translocator (ARNT) and estrogen related receptor gamma (ESRRG) were also predicted in the WB muscle. These findings indicate that there are considerable differences in upstream regulators based on downstream protein expression observed in WB myopathy and in high FE PedM broilers that may provide additional insight into the etiology of WB myopathy.

## 1. Introduction

The incidence of wooden breast (WB) muscle myopathy has increased in the last several years, possibly in response to genetic selection for growth performance and a shift to heavier market weight birds [1,2,3]. There have been several reports recently that have investigated WB myopathy to identify genetic signatures and histological defects in the progression of the disease (e.g., [3,4,5,6,7,8,9]). Collectively, these studies point toward; (a) phlebitis and inflammation, (b) oxidative stress and metabolic dysfunction, (c) myofiber degeneration, (d) lipid deposition and (e) development of hard pectoral muscle, particularly in the cranial region. All of these factors contribute to the poor muscle quality and lower shelf life of processed breast muscle fillets [10].

We have conducted a shotgun proteomic analysis on breast muscle tissue from pedigree male (PedM) broilers phenotyped for feed efficiency (FE) [11] and from commercial broilers with and without the presence of severe WB muscle myopathy [2]. It has been hypothesized that the increased incidence of WB might be related to the increased age at market weight as well as faster growth rates in broilers. Thus, there is a concern that increased selection for growth performance might be contributing to the increased incidence of WB myopathy. While WB is an area with unknowns, damaged mitochondria, low capillarity, lactic acidosis and hypoxia are all suggestive of an aerobic energy supply issue. Ingenuity Pathway Analysis (IPA) software was used to facilitate the organization and interpretation of the proteomic data in the studies by Kong et al. [11] and Kuttappan et al. [2]. A powerful feature of the IPA program is the ability to compare datasets using an overlay function in which data from one study can be projected (overlaid) onto another dataset to reveal similarities and differences between the two datasets. It can also be used to connect upstream regulators to diseases and functions. This overlay function can produce hypotheses for future studies and insight into fundamental mechanisms between these global expression datasets. Therefore, the intent of the current study is, firstly, to conduct upstream regulator analysis of WB data not previously reported in Kuttappan et al. [2], and secondly, to conduct comparisons of the WB data with the proteomic data associated with feed efficiency reported by Kong et al. [11].

## 2. Materials and Methods

### 2.1. Ethics Statement

The present study was conducted in accordance with the recommendations in the guide for the care and use of laboratory animals of the National Institutes of Health. All procedures for animal care complied with the University of Arkansas Institutional Animal Care and Use Committee (IACUC): Protocol Nos. 14,012 and 17,080.

### 2.2. Wooden Breast Myopathy Samples in Commercial Broilers

Breast muscle samples for wooden breast (WB) proteomics were obtained from male broilers (*n* = 24) that were randomly selected from a floor pen study at 52 d in a previous study [2]. After weighing, the birds were euthanized using carbon dioxide. The skin over the *Pectoralis major* muscle was excised and the muscle scored for woody breast [12]. Breast muscle samples (~5 g from right cranial region of the *Pectoralis major)* were collected, immediately snap frozen, and stored at −80 °C. Based on the lesions scores from histological analysis conducted in the same region, the muscle samples were categorized for severity of muscle lesions. From the 24 muscle samples, five samples were chosen that exhibited normal (NORM) histology and five others were chosen with severe (SEV) amounts of lesions to be subjected to proteomic analysis [2].

### 2.3. Breast Muscle Samples in Pedigree Male Broilers

Shotgun proteomics was conducted on breast muscle samples obtained from Pedigree Male (PedM) broilers individually phenotyped for high or low feed efficiency (FE) (*n* = 4 per group) [11]. These samples had been obtained from the right *Pectoralis muscle* in PedM broilers between 8–9 weeks that were humanely killed as part of a larger study [13].

### 2.4. Protein Extraction

In Kong et al. [11], muscle was homogenized in 1.5 mL of 20 mM potassium phosphate buffer at pH 7.4 using a hand-held Tissue-Tearor (Biospec Products Inc., Bartlesville, OK, USA) at speeds varying from 5000 rpm to 32,000 rpm. Following homogenization, samples were centrifuged at 10,000× *g*, and the supernatant was collected. In Kuttappan et al. [2], protein samples were precipitated using TCA-acetone with 8 M urea, 100 mM Tris HCl (pH 8.5) with 5 mM Tris 2-carboxyethyl phosphine at room temperature. Following dissolution and reduction, a 1/20th volume of 200 mM iodoacetamide was added and alkylation carried out for 15 min in the dark at room temperature. The sample was then diluted with four volumes of 100 mM TrisHCl, and digested with trypsin overnight at 37 °C. Protein concentrations in the supernatant in both studies were determined using the Bicinchoninic Acid Protein Assay (Sigma Aldrich, St. Louis, MO, USA). NORM and SEV muscle samples were then diluted to a protein concentration of 20 μg/150 μL in phosphate buffer, and the samples were stored at −80 °C until further analysis.

### 2.5. Shotgun Proteomics

Individual extracted proteins were used in shotgun proteomics analysis with in-gel trypsin digestion followed by tandem mass spectrometry (MS/MS) conducted at the University of Arkansas for Medical Science proteomics core lab (UAMS, Little Rock, AR, USA) [11]. Raw data were analyzed by database searching using Masco (Matrix Science, Boston, MA, USA) and UniProtKB database (http://www.uniprot.org/help/uniprotkb) with the result compiled using the Scaffold Program (Proteome Software, Portland, OR, USA).

In Kuttappan et al. [2], digested sample homogenates were acidified with 1% formic acid, and purified by reversed phase chromatography using C18 affinity media (Omix-Agilent, Santa Clara, CA, USA). Each sample was subjected to three replicate analyses for LC-MS/MS using a hybrid-OrbitrapXL mass spectrometer (ThermoFisher Scientific, Waltham, MA, USA) according to Voruganti et al. [14]. Mass spectrometry analysis was conducted in the DNA/Protein Resource Facility (Oklahoma State University, Stillwater, OK, USA).

### 2.6. Upstream Regulator Analysis

Upstream regulator analysis by IPA is based on: (a) the number and degree of differential expression of downstream target molecules in the existing dataset and (b) prior knowledge of relationships between upstream transcriptional regulators and their downstream target molecules obtained in published literature citations that have been curated and stored in the IPA program. Upstream regulator analysis determines how many known targets or regulators are within the user′s dataset, and compares each differentially expressed molecule to the reported relationship in the literature. If the observed direction of change is mostly consistent with either activation or inhibition of the transcriptional regulator, then a prediction is made and an activation z score generated that is also based on literature-derived regulation direction (i.e., “activating” or “inhibiting”). Activation z scores >2.0 indicate that a molecule is activated whereas activation z scores of <−2.0 indicate that a target molecule is inhibited. Qualified predictions can also be made with activation z scores between 2.0 and −2.0. The *p*-value of overlap measures whether there is a statistically significant overlap between the dataset molecules and those regulated by an upstream regulator is calculated using Fisher’s Exact Test, and significance is attributed to *p*-values < 0.05.

## 3. Results and Discussion

The main focus of this study is on upstream regulator analysis of severe-fulminant WB myopathy. However, as parts of the results and discussion below are concerned with comparing the WB myopathy proteomic data to a dataset obtained in PedM broilers phenotyped for high and low FE, a description of proteomics methods and of the animals from which samples were obtained for the two datasets is warranted.

### 3.1. Phenotypic Data for Birds in Wooden Breast and Feed Efficiency Studies

Body weights, wooden breast scores for commercial broilers and growth performance during phenotyping for feed efficiency in PedM broilers from which muscle samples were obtained for proteomic studies are presented in Table 1. It can be seen that there were significant differences in body weight and wooden breast (WB) myopathy scores in commercial broilers in severe vs. normal muscle. Additionally, although there were no differences in body weights in PedM broilers, there was a significant difference in feed efficiency due to lower gain while consuming the same amount of feed in the low FE PedM phenotype.

### 3.2. Proteomic Dataset Comparison

Proteomics conducted on high and low FE PedM phenotypes in Kong et al. [11] had been obtained eight years earlier in Bottje et al. [13] than for the WB myopathy study by Kuttappan et al. [2]. The timing of sample collection is important for at least two reasons; (a) it was well before WB myopathy had been observed in commercial broilers and (b) the genetic impact of selection of the PedM broiler line would have had sufficient time in the genetic pipeline to reach the level of commercial broilers in the study by Kuttappan et al. [2]. Breast muscle samples were obtained at 52 day in the WB study and between 56 and 63 day (after FE phenotyping 6–7 weeks) in the PedM FE study. It is worth noting that the number of samples in each study are small (*n* = 4 or 5), and therefore, this paper may run the risk of not being fully representative of the two populations in each study. However, it can also be noted that the difference in phenotypes were highly significant (*p* = 0.003 and *p* < 0.0001 for the WB and FE phenotypes, respectively), suggesting that the relative differences in global expression patterns could be considered representative of a larger phenotype population.

Proteomic analysis of the two studies were conducted at different laboratories. For the commercial broiler study [2], LC-MS/MS analysis was conducted using a hybrid LTQ-OrbitrapXL mass spectrometer (ThermoFisher Scientific, Waltham, MA, USA) as described previously [14], but using 40-cm C18 columns developed over a 2-h period with 0 to 40% acetonitrile. In the PedM broiler study [11], extracted individual proteins were subjected to shotgun proteomics analysis by in-gel trypsin digestion and tandem mass spectrometry (MS/MS) at the University of Arkansas Medical Science (UAMS) Proteomics Core Lab (Little Rock, AR). Both studies used the *Gallus* reference proteome downloaded from the UniProtKB (http://www.uniprot.org/help/uniprotkb) database to identify individual proteins from the spectrometric data. The annotation of proteins upstream regulator analysis and downstream functions were conducted using Ingenuity Pathway Analysis (IPA, Qiagen, Redwood City, CA USA).

### 3.3. Upstream Regulators and Functional Analysis

Figure 1 provides a visual guide to assist in understanding subsequent figures and tables in this study. The predictions of activation (orange background, positive activation Z score) or inhibition (blue background, negative activation Z score) of upstream regulators (Table 2) and functions (Table 3) in WB muscle were determined using the Ingenuity Pathway Analysis program. These predictions (activation Z scores) were based on: (1) differential protein expression in the dataset and (2) relationships of upstream regulators and downstream target molecules in the IPA scientific literature database.

Appendix A contains all of the differentially expressed proteins in the WB myopathy datasets that will appear in tables and figures below. Proteins highlighted in green were down-regulated and those in red were upregulated in the WB myopathy muscle, respectively.

In Table 2 and Table 3, proteins in red and green (underlined) lettering denote up- and down-regulation, respectively, in WB myopathy. The protein abbreviations in Table 2 and Table 3 are defined in Appendix A along with the *p* value and fold difference in expression for each protein. Using an overlay function in the IPA program, it is possible to compare the expression of proteins and predictions of upstream regulators and functions in different datasets. Thus, a qualitative comparison of the WB myopathy proteomics data in the present study can be made to the high vs. low FE proteomics data from Kong et al. [11].

Examples of functional networks in Table 2 are presented in Figure 2. In the vasculogenesis function network (Figure 2A), all up-regulated proteins, with the exception of CALR (calreticulin), contributed to the prediction of activation (activation *Z* score = 2.43, *p* = 4.46 × 10^−4^). The yellow dashed inhibitory line (indicating that the relationships of up-regulation of CALR and predicted activation of vasculogenesis are inconsistent with literature citations) for CALR indicates that enhanced vasculogenesis is typically associated with down-regulation of CALR. The down-regulation of ACP1 would contribute to the predicted activation of vasculogenesis because it would reduce the inhibitory effect on this function. On the other hand, the effects of PKM and LDHA were inconsistent with their down-regulation, indicating that increased expression of these proteins would normally be associated with increased vasculogenesis.

It is worth noting that predictions for ANGPT2 (described in more detail below) or vasculogenesis were not made in the high vs. low FE proteomic dataset; cell death of muscle cells, necrosis of muscle and apoptosis were all predicted to be inhibited in the high FE phenotype. The activation state and *p* value of overlap were all significant for necrosis of muscle cells (−2.19, *p* = 1.52 × 10^−5^), cell death of muscle cells (−2.59, *p* = 1.54 × 10^−6^), and apoptosis of muscles (−2.81, *p* = 1.29 × 10^−4^) for the high FE phenotype, which suggests that a commonality of mechanisms were presented in the high FE PedM broiler and for broilers exhibiting WB myopathy.

Using the overlay function of the IPA program, it is possible to project one dataset onto another to see commonalities and differences between the datasets. In Figure 2B, the FE proteomic dataset (from Kong et al. [11]) was projected (overlaid) onto the WB myopathy vasculogenesis functional network. The result shows all of the proteins in WB proteomics dataset were not differentially expressed (fold difference, 1.3, *p* < 0.05) in the FE dataset, i.e., vasculogenesis was not predicted to be a significant function in the high vs. low FE PedM phenotypes.

In the necrosis functional network (Figure 3), all up- and down-regulated proteins in WB myopathy contributed to the prediction of inhibition of necrosis; there were no inconsistencies in the relationships between the proteins and the downstream effect of predicted inhibition of necrosis. With the large number of predictions that were obtained in the WB myopathy dataset, the discussion of each upstream regulator may not be sufficient in detail and critical literature citations may be missed. Nonetheless, we hope it will spark hypotheses to be tested that will provide insight into this disease that can lead to preventive measures to ameliorate significant economic losses to the poultry industry.

Although there was no commonality between WB myopathy and FE datasets using the overlay function in IPA, three functions—muscle cell death, apoptosis of muscle cells, and necrosis of muscle cells—were predicted to be inhibited in the high FE vs. low FE PedM dataset (Appendix A). Thus, there was a commonality between WB myopathy and high FE with respect to these functions. This will be discussed in a little more detail in the summary section towards the end of this manuscript.

### 3.4. Upstream Regulators

#### 3.4.1. Angiopoeitin 2

The top predicted activated upstream regulator in the WB myopathy was ANGPT2 (Table 2). Murtyn et al. [15] indicated that hypoxia and circulatory insufficiency may play a role in the development of WB myopathy. Thus, the predicted activation of ANGPT2 by IPA could be a tissue-mediated signal to increase blood vessel development in damaged tissue to enhance oxygen delivery. This hypothesis is supported by the predicted activation of functions of development of vasculature, vasculogenesis and angiogenesis shown in Table 3. Moffarri and Hossein [16] reported that ANGPT2 expression in primary skeletal muscle myocytes was responsive to hydrogen peroxide-induced oxidative stress, but not to proinflammatory cytokines. As ANGPT2 promotes skeletal myoblast survival and repair, Moffarri and Hossein [16] hypothesized that muscle-derived ANGPT2 production plays a positive role in muscle fiber repair. Conversely, ANGPT2 activation could help in the delivery of oxygen and nutrients to aid growth and muscle development as birds with WB myopathy were heavier than birds without WB myopathy (Table 1). In line with this general connection of WB to impaired aerobic energy supply, we have recently found that heavier birds (known to be pre-disposed to WB) possess a lower skeletal muscle mitochondrial content, which would reduce the capacity for aerobic ATP synthesis [17].

The ANGPT2 network of downstream molecules presented in Figure 4A indicates that all of the downstream molecules in WB myopathy were up-regulated compared to normal muscle. Each of these molecules are indicated to be up-regulated indirectly (dashed orange line and arrow) by ANGPT2, with the exception of ribosomal protein 6 (RP6), whose up-regulation was not predicted (black dashed arrow line). This relationship, i.e., an unpredicted effect, occurs when there are too few literature citations in the IPA database to establish a clear relationship between the upstream regulator and the downstream target molecule. When the FE proteomics dataset was overlaid (projected) onto the WB myopathy ANGPT2 network (Figure 4B), there were no commonalities in protein expression and therefore no prediction could be made of ANGPT2 activity.

Using the ′Grow′ function of the IPA program, it is possible to link upstream regulators to downstream functions, as shown in Figure 4C. In this figure, predictions of activation of vasculogenesis and angiogenesis in WB myopathy are the result of increased VIM, CRYAB, HSPA5 and HSP90AA1 expression and the predicted activation of ANGPT2. All of these are linked by dashed orange lines with arrows. The dashed lines indicate that at least one additional step was between the upstream molecule and the downstream target. Apoptosis and necrosis were predicted to be inhibited in WB myopathy due to increased expression of CRYAB, HSPA5, HSP90AA1, CALR and HSPA2 and by the predicted activation of ANGPT2. These relationships to necrosis and apoptosis functions are indicated by dashed blue lines with a short perpendicular line at the end indicating inhibition. The lighter blue color for necrosis indicates that the prediction made for this function, based on proteins in this network, was qualitatively less than for apoptosis. The darker color (stronger qualitative prediction) for apoptosis is due in part to more proteins in the regulatory network (five up-regulated proteins plus ANGT2) compared to four proteins in the necrosis network.

#### 3.4.2. Epidermal Growth Factor Receptor

The upstream regulator network of epidermal growth factor receptor (EGFR) in WB myopathy is presented in Figure 5. EGFR was predicted to be activated in WB myopathy, but the overlay of the FE proteomics dataset resulted in no prediction, indicating no commonality with this upstream regulatory network between WB myopathy and high FE. There are thousands of literature citations for EGFR involvement in carcinoma, but here we will focus on a few citations that are pertinent to EGFR in skeletal muscle development. Olwin and Hauschka [18] reported that EGFR and fibroblast growth factor receptor were permanently lost in terminal differentiation of adult mouse skeletal muscle in vitro. It was also reported that whereas EGF increases in skeletal muscle as pigs age, mRNA expression of the receptor declines [19]. Finally, Leroy et al. [20] reported that down-regulation of EGFR triggers differentiation of human myoblasts in cell culture, indicating that EGFR expression normally declines with age. Therefore, the prediction of activation of EGFR in WB myopathy might indicate an abnormal process that contributes to the pathophysiology in WB myopathy.

#### 3.4.3. Transforming Growth Factor Beta 1 (TGFB1)

TGFB1 was predicted to be activated in WB myopathy proteomics (Table 2, Figure 6A) as well as in the high FE PedM phenotype (Figure 6B). While the TGFB1 networks in the two studies did not share any differentially expressed proteins, the overlay of the FE proteomics data on the WB dataset (Figure 6C) as well as the overlay of the WB proteomics data on the FE dataset (Figure 6D) resulted in predictions of the activation of TGFB1. TGFB1 is a cytokine that is involved in controlling the growth, proliferation and differentiation of cells and able to serve in an autocrine manner to regulate its own expression [21]. This autocrine stimulation is indicated by the semi-circular arrow in each of the networks in Figure 6. In the study conducted by Hubert et al. [22], TGFB1 was also identified as one of the upstream regulators of the WB myopathy. The role of TGFB1 in this myopathy may be explained in part by its effects on cell differentiation and mitochondrial function. Under normal conditions, this cytokine is involved in the regulation of cell differentiation, increased mitochondrial activity and increased oxidative phosphorylation. However, studies have shown that increased TGFB1 expression is detrimental and can produce mitochondrial dysfunction with increased oxygen radical production that compromises the mitochondrial antioxidant system [23,24,25]. The resulting mitochondrial perturbations have been illustrated to lead to the inhibition of myogenic differentiation and the formation of tissue fibrosis [26,27]. The potential for diminished mitochondrial biogenesis in WB myopathy is further supported by predicted inhibition of PGC1α, which is discussed in more detail below. Feed efficiency studies in cattle have also identified TGFB1 as being an upstream regulator [28,29]. This provides further evidence of the of TGFB1 involvement in genetic differences in the various feed efficiency phenotypes.

Protein abbreviations for WB myopathy data (A, C) are defined and differential expression of proteins are provided in Appendix A. Protein abbreviations for FE proteomics (B and D) are as follows: ACTA1 (actin, alpha 1, skeletal muscle), ACTC1(actin, alpha, cardiac muscle 1), ALB (albumin), ARF (ADP-ribosylation factor 1), ARF4 (ADP-ribosylation factor 4), C4BPA (complement component 4 binding protein, alpha), CAV1 (caveolin 1), CCT2 (cell division cycle 42), CKM (creatine kinase, muscle), CSPG4 (chondroitin sulfate proteoglycan 4), CTSC (cathepsin C), EEF1A (eukaryotic translation elongation factor 1 alpha 1), ENO2 (enolase 2, gamma neuronal), GPI (glucose 6 phosphate isomerase), IRS1 (insulin receptor S1), OARD1 (O-acyl-ADP-ribose deacylase), PRPS1 (phosphoribosyl pyrophosphate synthetase 1), PSMD1 (proteosome 26S subunit, non-ATPase, 1), SLC25A4 (solute carrier family 25 (mitochondrial carrier; adenine nucleotide translocator), member 4), VDAC2 (voltage-dependent anion channel 2).

Linkage of TGFB1 activation to downstream functions through the differentially expressed proteins is provided in Figure 7. For simplicity, only relationships that clearly predicted activation (angiogenesis, vasculogenesis) or inhibition (apoptosis, necrosis) are presented in this figure. The expression of eight proteins in this regulatory network (PGK1, CTSB, VIM, HSP90AA1, FLNA, HSPA5, ANAXA2 and RHOA) were involved with predictions of increased angiogenesis and vasculogenesis in WB myopathy. The up-regulation of all the proteins, with the exception of PGK1, which was down-regulated, were downstream targets of TGFB1 that contributed to the predictions of enhanced blood vessel-capillary formation. Eight up-regulated proteins (HSP90AAA1, FLNA, HSPA5, ANXA2, RHOA, RACK1, NFRNPH1 and DES) and one down-regulated protein (HINT1) were involved in the predictions of reduced apoptosis and necrosis in wooden breast myopathy.

#### 3.4.4. Nuclear Factor (Erythroid-Derived 2)-Like 2 (NFE2L2)

Nuclear factor (erythroid-derived 2)-like 2 (NFE2L2) coordinates cellular response to oxidative stress [30,31]. NFE2L2 was predicted to be activated in both WB proteomics and in the FE proteomics dataset in PedM broilers (Figure 8A,B), but the predictions were based on different downstream molecules. Overlaying of each dataset on the other upstream regulatory network (Figure 6C,D) resulted in predicted activation of NFE2L2.

Under normal (non-oxidant stress) conditions, the NFE2L2 protein remains bound to Cullin3 (CUL3) and Kelch like-ECH protein 1 (KEAP1), and is then rapidly directed (within 15–20 min) to proteasomes for degradation [32]. However, in response to oxidative stress, NFE2L2 is transported to the nucleus, where it stimulates antioxidant gene expression after binding to the antioxidant response element. Recently, NFE2L2 was predicted to be activated in high feed efficiency (HFE) compared to low feed efficiency (LFE) broiler phenotypes based on downstream target molecules [11,33]. It has been shown that NFE2L2 undergoes post-transcriptional and translational modifications that can affect the amount of NFE2L2 protein produced, which, along with rapid activation or degradation, makes it an extremely labile protein [32,34]. Avian NFE2L2 appears to function the same as mammalian NFE2L2, but since it shares only 67% homology with mammalian NFE2L2 [35], it may have other roles in avian tissues besides coordination of antioxidant response. In a recent review, NFE2L2 was observed to affect mitochondrial function in many ways, including efficiency of oxidative phosphorylation, mitochondrial biogenesis and mitochondrial integrity [36].

#### 3.4.5. X-Box Binding Protein 1

X-box binding protein 1 (XBP1) was predicted to be activated in WB myopathy (Table 2). There was no commonality of protein expression of this upstream regulator in the FE proteomic dataset. Myoblasts that were forced to differentiate in response to XBP1 overexpression produced myotubes that were shorter and less mature indicative of impaired differentiation relative to control cells [37]. Of particular potential relevance to WB myopathy is the following from this study: “Our observations suggest that skeletal muscle tissue is particularly sensitive to physiological stressors that trigger the unfolded protein response, namely glucose deprivation, anabolic stimulation, and perhaps other endoplasmic reticulum stresses, such as hypoxia, imbalances in calcium homeostasis, and ischemia. These perturbations could be triggered by normal or pathological states.” The presence of hypoxia in WB myopathy has been reported [38] and will be discussed in greater detail below.

#### 3.4.6. Rapamycin-Insensitive Companion of Mammalian Target of Rapamycin (RICTOR)

RICTOR plays an important role in actin-cytoskeletal development, as formation of this structure was impaired in a RICTOR knockout mouse model [39,40,41]. Ablation of RICTOR (mTORC2) did not adversely affect muscle function in mice, results that contrast dramatically with the loss of function and development of muscle dystrophy when RAPTOR (mTORC1) was removed [42].

RICTOR was predicted to be inhibited in WB myopathy with activation z scores of −3.61 (Table 2). RICTOR was also predicted to be inhibited in PedM broilers exhibiting high FE [11]. The upstream regulator networks of RICTOR in the current WB myopathy study and in the FE proteomics study are presented in Figure 9A,B, respectively. Although both studies yielded predictions of RICTOR inhibition, it is apparent that the predictions were based on different downstream target molecules in the respective datasets. The overlay of the datasets on each other yielded no prediction giving indication of no commonality in the RICTOR networks. Thus, this appears to indicate that the predictions of inhibition of RICTOR in WB myopathy and FE breast muscle proteomics are due to separate mechanisms. The connection between inhibition of RICTOR to inhibition of necrosis shown in Figure 9C is shown to be mediated by up-regulation of several ribosomal proteins. Apoptosis was predicted to be weakly activated in this network through combined inhibition of RICTOR and up-regulation of RPS3.

#### 3.4.7. Peroxisome Proliferator-Activating Gamma Coactivator 1 Alpha (PPARGC1α or PGC1α)

Because PGC1α has been described as the master regulator of mitochondrial biogenesis (Nisoli et al. [43,44], the predicted inhibition of PGC1α by upstream regulator analysis (Figure 10A, activation Z score = −2.35, *p* = 1.92 × 10^−5^ suggests that mitochondrial biogenesis, and possibly mitochondrial function, would be impaired in WB myopathy. In contrast, the upstream regulator analysis of PGC1α was predicted to be activated in the high FE PedM broiler phenotype (Figure 10B, activation Z score = 2.45, *p* = 1.84 × 10^−4^). There was no commonality of protein expression between the WB and FE datasets. The mitochondrial canonical pathways in WB and FE proteomics datasets are shown in Figure 10C,D, respectively. Whereas Complex I of the mitochondrial electron transport chain was predicted to be inhibited in WB myopathy (Figure 10C), Complex I, III and IV were predicted to be activated based on expression of proteins in the FE dataset (Figure 10D). Complex I (NADH dehydrogenase), which accepts NADH-linked energy substrates, is the largest complex in the electron transport chain, consisting of approximately 40 proteins. Thus, inhibition of this complex could have a dramatic influence on the energy production in WB tissues. The potential for diminished mitochondrial function in WB myopathy is supported by a recent report showing increased mitochondrial damage and mitophagy in histological analysis of WB myopathy [45]. Several studies provide evidence of enhanced mitochondrial function and complex activities in the high FE PedM phenotype (e.g., see review by [46]) that may be attributed to increased activity of PGC1α in the high FE phenotype shown in Figure 10B.

The regulatory network for PGC1α presented in Figure 10E indicates that, unlike the overall prediction of vasculogenesis activation (Table 3), vasculogenesis was predicted to be inhibited by a combination of down-regulation of LDHA and predicted inhibition of PGC1α. Reactive oxygen species (ROS) production was predicted to be enhanced in WB myopathy due to a combination of LDHA down-regulation and up-regulation of C3. Enhanced ROS production may contribute to tissue damage reported in WB myopathy. In this network, oxidation of fatty acid was predicted to be inhibited in WB myopathy in the present study due to the up-regulation of C3, down-regulation of GOT2 and predicted inhibition of PGC1α and the quantity of adipose tissue was predicted to be activated in WB myopathy. These findings agree with many studies, indicating that lipid accumulation in muscle is a characteristic in WB myopathy [6,47,48].

#### 3.4.8. Aryl Hydrocarbon Receptor Nuclear Translocator (ARNT) and Aryl Hydrocarbon Receptor Nuclear (AHR)

ARNT and AHR were predicted to be inhibited in WB myopathy (Table 2). The regulatory networks shown in Figure 11A,B, respectively. for ARNT indicate that fibrosis would be predicted to be enhanced, and glycolysis, vasculogenesis as well as apoptosis would be predicted to be inhibited based on expression of proteins within this regulatory network. In contrast, angiogenesis and necrosis were predicted to be activated in the AHR regulatory network.

Murtyn et al. [15] provided a sound argument for the presence of hypoxic conditions playing a role in WB myopathy that was based on the differential expression of several downstream molecules that are responsive to hypoxia inducible factor 1 (HIF1 and HIF-dependent expression). However, they also indicated that transcripts of the two subunits of HIF-1, HIF-1α and ARNT were not differentially expressed between normal and WB myopathy breast muscle. Since both AHR and ARNT are involved in other cellular processes (e.g., xenobiotic metabolism), the predictions of inhibition in the present study (and lack of difference in gene expression in Murtyn et al. [15]) may have nothing to do with tissue hypoxia, but rather, are playing roles in other cellular processes. For example, AHR knockout in aortic smooth muscle cells resulted in the deregulation of components of TGF-β signaling and could directly affect the metabolism of toxic substances [49]. Recent reviews indicated that there are numerous functions that AHR may play in the cell [50] and immunity [51]. Greene et al. [38] measured lower oxygen levels in breast muscle tissue exhibiting WB myopathy compared to normal tissue. Furthermore, several genes associated with hemoglobin in red blood cells and HIF-1α in breast muscle were down-regulated in WB compared to normal breast muscle tissue [38].

Some discussion is warranted with regard to the predicted inhibition of AHR in the present study. First, predictions of activation or inhibition of an upstream molecule generated in the IPA program are based on the expression of downstream molecules specifically associated with the dataset being examined. In the case of the WB proteomic dataset in the present study, there were far fewer differentially expressed proteins (~130) that were detected compared to approximately 1500 differentially expressed transcripts detected by Murtyn et al. (2015) [15]. Secondly, denaturing gels used in separating proteins prior to protein expression analysis can result in a loss of many proteins with large hydrophobic regions (e.g., membrane-bound or membrane associated proteins), which in turn could influence the predictions generated by the IPA program. Third, the lack of differential expression of ARNT and HIF-1α m-RNA reported by Murtyn et al. [15] does not mean that the proteins would also not be differentially expressed. Furthermore, these proteins may also undergo post-translational modifications that would be required for activation of downstream target molecules. The predicted activation of ANGPT2 in WB myopathy could be a response to tissue hypoxia to increase vascularization caused by tissue damage and/or hypoxia. Thus, further examination AHR and ARNT signaling in WB myopathy is warranted.

Although no predictions were made regarding AHR and ARNT in the FE proteomics study by Kong et al. [11], ESR1 (estrogen receptor 1) was predicted to be activated (activation Z score = 1.97, 3.93 × 10^−3^). In the IPA gene view in the Qiagen software, ESR1 is indicated to be a member of AHR-aryl hydrocarbon-Arnt-ESR1. Thus, this would appear a major difference between the WB and FE proteomics datasets.

#### 3.4.9. Estrogen Regulator Receptor Gamma (ERSSγ)

ERSSγ was predicted to be inhibited in WB myopathy (Table 2). The regulatory network of ERSSγ shown in Figure 12 indicates that vasculogenesis would be inhibited by the down-regulation of LDHA and PKM, whereas down-regulation of GAPDH caused a prediction of enhanced necroptosis in WB myopathy. Again, both of these predictions are opposite to functions outlined in Table 3, with the discrepancy attributed to the larger number of proteins involved in the overall predictions shown in Table 3.

Fan et al. [52] reported that ESRRγ helps in reducing muscle damage and improves muscle function in a double PGC1α/β knockout mouse model. The loss of PGC1α/β resulted in a decrease in expression of myoglobin and a lighter color of muscle consistent with a change from oxidative to glycolytic fibers, which were largely restored when ESRRγ transgenic mice were crossed with the PGC1α/β knockout mice [52]. Defects in mitochondrial function from the PGC1 α/β knockout were also restored by crossing with the ESRRγ transgenic mice. Furthermore, ESRRγ also increased the expression of genes associated with angiogenesis (e.g., vascular endothelial growth factor A and fibroblast growth factor 1) and vascular density (CD31).

### 3.5. Summary and Synopsis

There were two major goals of this study; (1) to present an upstream regulatory factor analysis associated with wooden breast myopathy to expand our understanding of this disease and (2) to determine where contributions to WB myopathy might be linked to selection for performance (high FE) in PedM broilers. Table 4 summarizes the results of comparison of upstream regulator analysis between the WB proteomic data to the FE proteomic data reported by [11]. With the exception of TGFB1 and NFE2L2, there were no commonalities for upstream regulators between the WB and FE proteomics data. While PPARGC1α was predicted to be inhibited in WB myopathy, this upstream regulator was predicted to be activated (based on a completely different set of differentially expressed proteins) in muscle of the high FE compared to the low FE PedM broiler. From these findings, we hypothesize that mitochondrial biogenesis would be inhibited in WB myopathy and activated in the high FE PedM broiler [11].

Figure 13 provides a graphical presentation of the upstream regulators that were predicted to be activated (orange) or inhibited (blue) in WB myopathy (13A) or FE proteomics (13B) and predicted interactions with other upstream regulators described in this study. This figure was generated by uploading the upstream regulator networks in Table 2, and then removing differentially expressed downstream target proteins to reveal predicted interactions between the upstream regulators. Major functions/processes, including the prediction of the activation of angiogenesis and the inhibition of cell death (apoptosis and necrosis), were discussed previously. The predicted activation of TGFB1 is indicated as having an indirect (through one or more mediators) inhibitory effect on PPARGC1A and predicted inhibition of PPARGC1A would directly inhibit EGFR. Predictions of reduced activity of AHR and ARNT both contribute to the prediction of lower ESRRG activity. Similarly, predicted inhibition of ARNT and AHR would inhibit ESRRG as well. Because PPARGC1A and ESRRG enhance mitochondrial biogenesis, their predicted inhibition could be hypothesized to lower mitochondrial biogenesis. Elevated NFE2L2 activity is predicted to result in enhanced activation of TGFB1. The predicted inhibition of AHR would function to enhance TGFB1 activity, since AHR has been shown to inhibit TGFB1. NFE2L2 activation would contribute to the predicted activation of TGFB1. Effects of ANGPT2 and TGFB1 would enhance EGFR and their combination would lead to the prediction of enhanced angiogenesis in WB myopathy.

Figure 13B presents upstream regulators from the FE proteomics data. No prediction was made concerning ANGPT2, and unlike WB myopathy, EGFR was predicted to be inhibited in the high FE phenotype. No prediction was made in high vs. low FE data concerning angiogenesis/vasculogenesis. TGFB1 was predicted to be activated in high FE, as were NFE2L2 and PPARGC1A, but as indicated in the text, these were based on different downstream target molecules compared to the WB myopathy data. No predictions were made for AHR, ESSRG or ARNT, but estrogen receptor 1 (ESR1), a member or the ARN family, was predicted to be activated in the high FE phenotype. The combination of increased PPARGC1A and ESR1 would act to enhanced mitochondrial biogenesis in the high FE phenotype.

All of the predictions presented in this study are just that—predictions. Each of these predictions of upstream regulator activity represents hypotheses and mechanistic studies that need to be conducted in order to determine if these upstream regulators contribute to the development of WB muscle myopathy. We hope that they will help shed light on the understanding of WB myopathy that will lead to methods, whether nutritional or chemical, that can be used to ameliorate WB muscle myopathy in commercial broilers.

## Figures and Tables

**Figure 1 foods-10-00104-f001:**
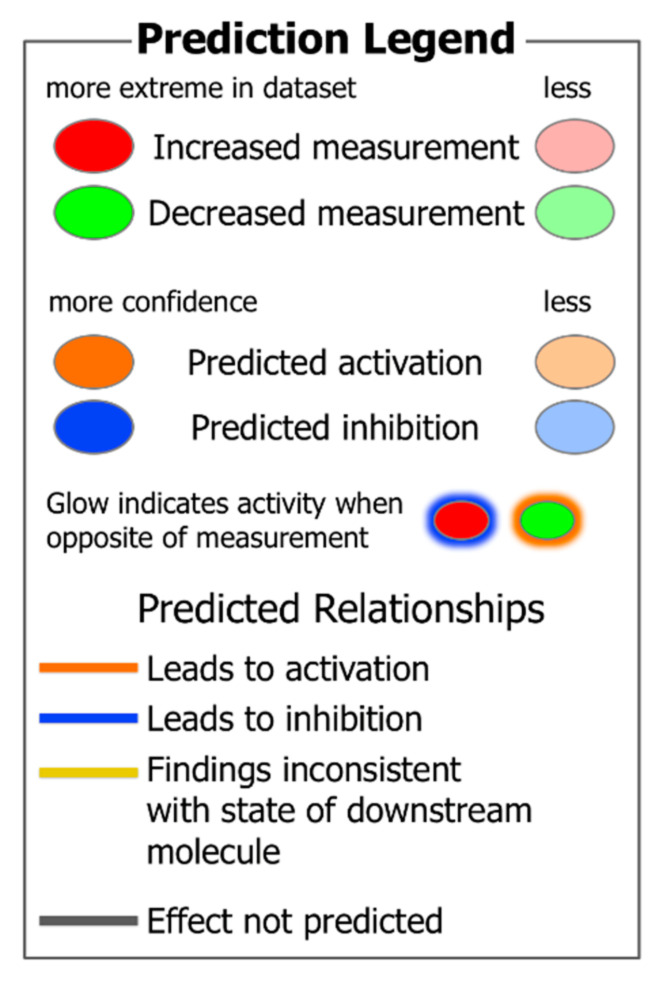
A prediction legend for the interpretation of protein expression and upstream regulator and function analysis in tables and figures.

**Figure 2 foods-10-00104-f002:**
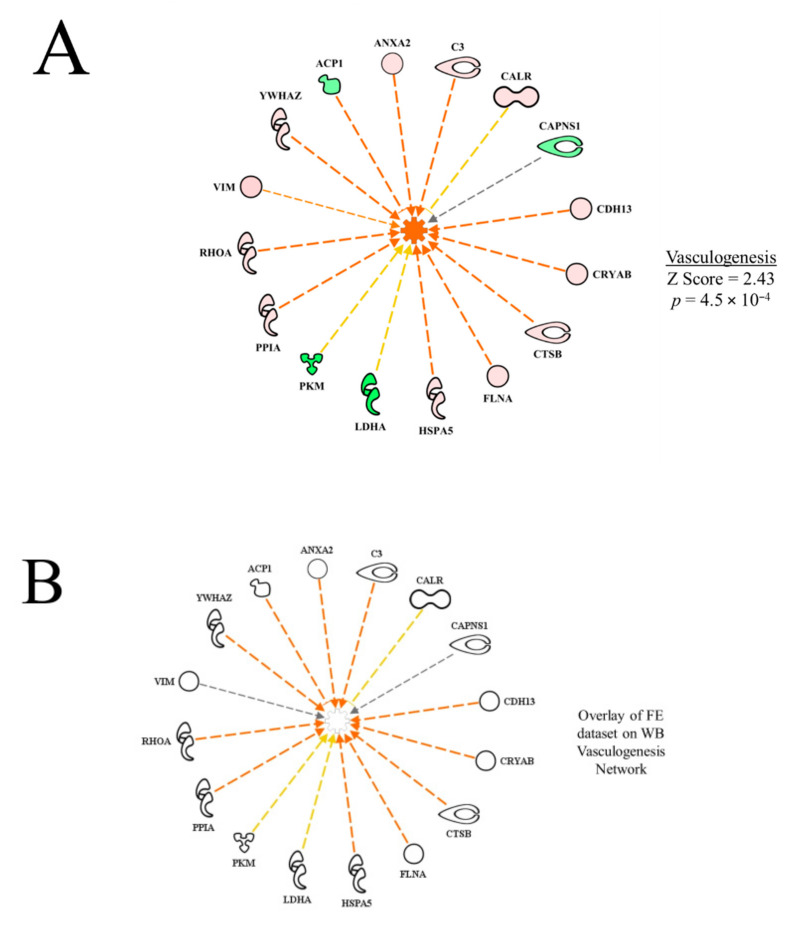
Network of differentially expressed proteins in the wooden breast (WB) myopathy dataset used in calculating the activation Z score for vasculogenesis (**A**). Overlay of the feed efficiency (FE) proteomic dataset from Kong et al. [11]; (**B**) indicates no commonality of differentially expressed proteins in WB vasculogenesis network Protein abbreviations are defined and differential expression of proteins are provided in Appendix A.

**Figure 3 foods-10-00104-f003:**
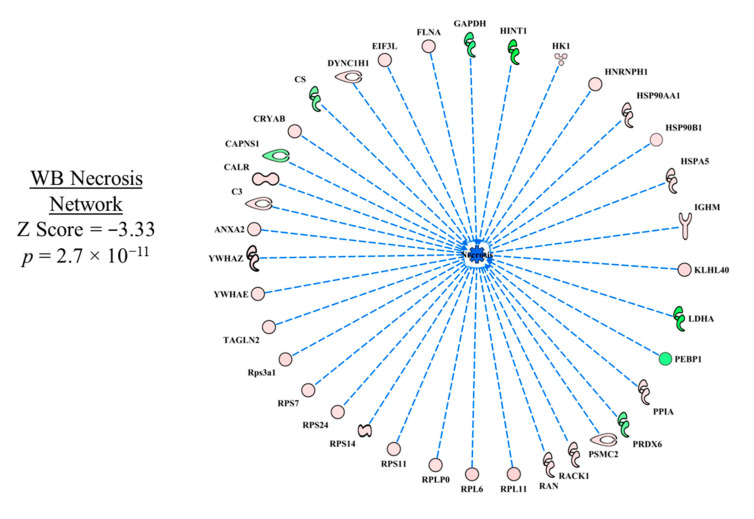
The network of differentially expressed proteins that was used to predict inhibition of necrosis in wooden breast (WB) myopathy. Protein abbreviations are defined and differential expression of proteins are provided in Appendix A.

**Figure 4 foods-10-00104-f004:**
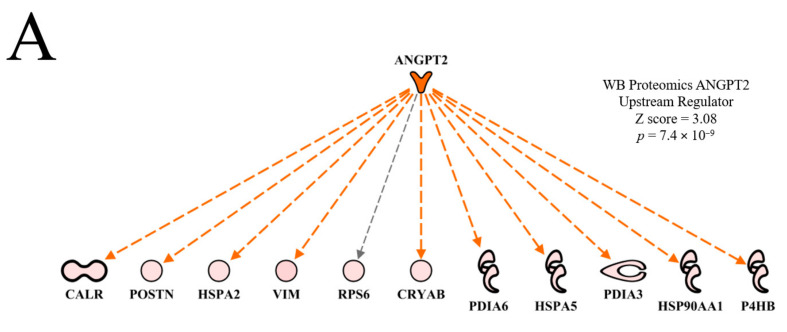
The upstream regulatory network of proteins displayed in hierarchical format for Angiopoeitin 2 (ANGPT2) used in the prediction of activation of this molecule (activation *Z* score and *p* value of overlap as shown in (**A**)). Overlay of the feed efficiency (FE) proteomics dataset from Kong et al. [11] reveals no commonality of differentially expressed proteins between the wooden breast (WB) and FE data (**B**). A regulatory network for ANGPT2 indicates target proteins in the dataset that contributed to the predictions of active vasculogenesis and angiogenesis, and inhibited necrosis and apoptosis is provided in (**C**). Protein abbreviations are defined and differential expression of proteins are provided in Appendix A.

**Figure 5 foods-10-00104-f005:**
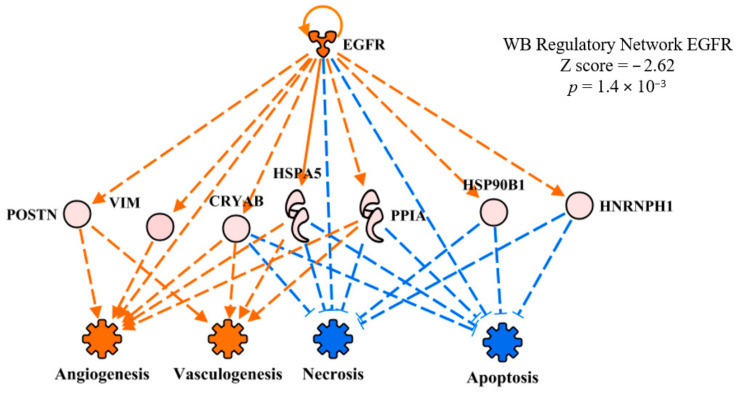
The regulatory network of epithelial growth factor receptor (EGFR) showing downstream target proteins contributing to the prediction of activation of vasculogenesis and inhibition of necrosis and apoptosis in wooden breast (WB) myopathy. Protein abbreviations are defined and differential expression of proteins are provided in Appendix A.

**Figure 6 foods-10-00104-f006:**
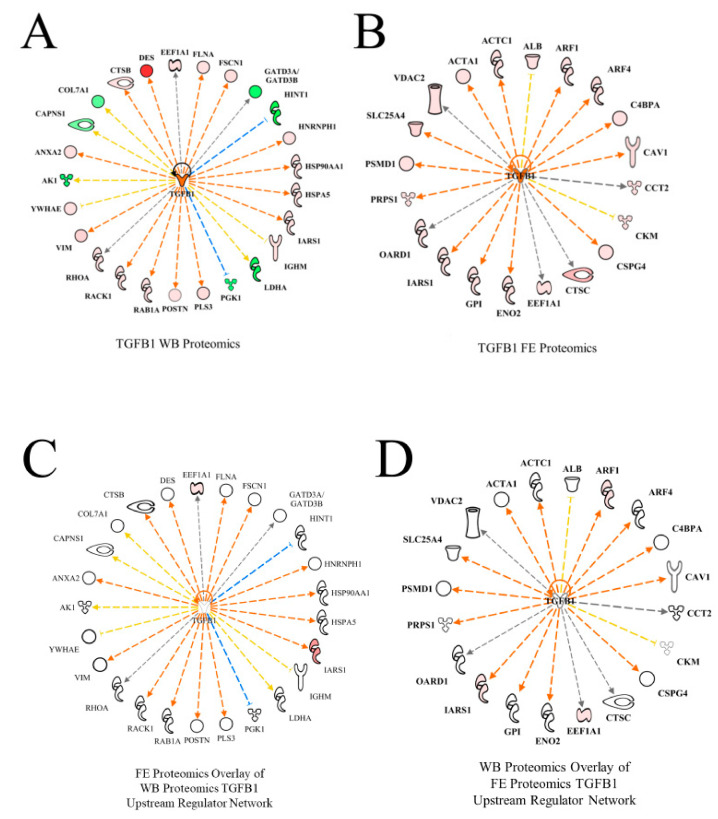
The upstream regulatory network of transforming growth factor beta 1 (TGFB1) for wooden breast (WB) (**A**) and feed efficiency (FE) (**B**) proteomic datasets. Downstream target proteins that were differentially expressed are shown as up-regulated in pink or red or down-regulated in WB myopathy and high FE PedM broilers. Overlay of the FE proteomic dataset on the WB myopathy network (**C**) and of the FE data on the WB network (**D**) resulted in predicted activations of TGFB1 as well. Therefore, despite dissimilar proteomic landscapes of the two datasets, TGFB1 is predicted to be active in both WB myopathy and high FE.

**Figure 7 foods-10-00104-f007:**
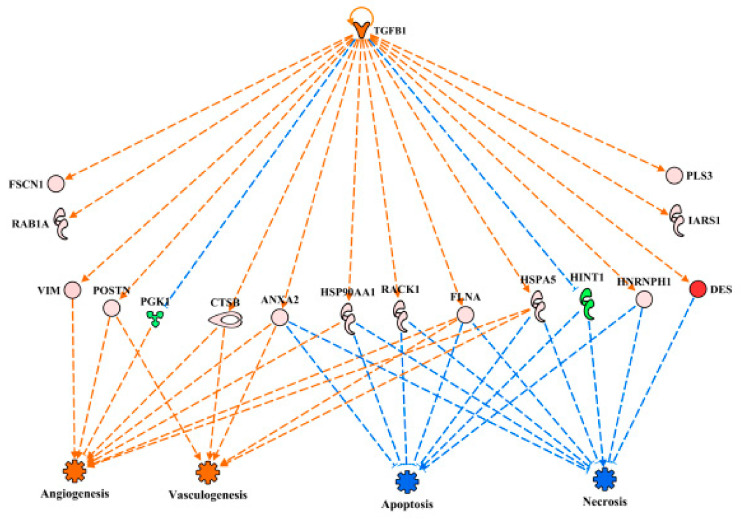
The regulatory network of TGFB1 in wooden breast (WB) myopathy leading to predicted activations of angiogenesis and vasculogenesis and inhibition of apoptosis and necrosis through downstream target proteins in WB myopathy. Protein abbreviations are defined and differential expression of proteins are provided in Appendix A.

**Figure 8 foods-10-00104-f008:**
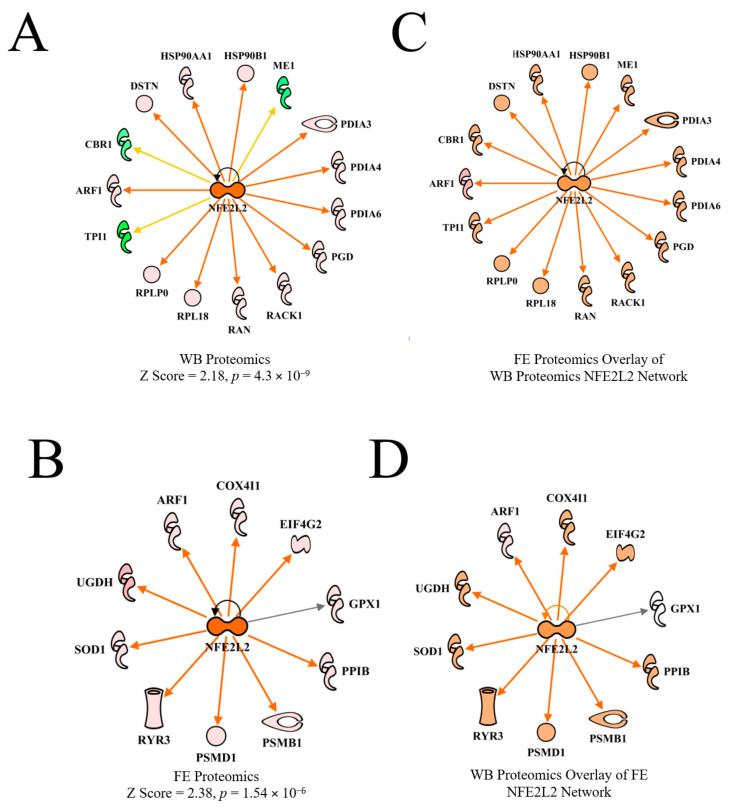
The upstream regulator network for NFE2L2 (nuclear factor (erythroid-derived 2)-like 2) for wooden breast (WB) myopathy (activation Z score = 2.18, 4.3 × 10^−9^) compared to normal muscle (**A**) and for the high vs. low PedM broiler feed efficiency (FE) proteomic data (activation Z score = 2.38, *p* = 1.54 × 10^−6^) (**B**). Overlay of the FE data on the WB myopathy data (**C**) and vice versa (**D**) both resulted in predicted activations of NFE2L2. Protein abbreviations in B and D in the FE data set are as follows: ARF1 (ADP-ribosylation factor 1), COX4I1 (cytochrome oxidase subunit IV isoform 1), EIF4G2 (eukaryotic translation initiation factor 4 gamma 2), GPX1 (glutathione peroxidase 1), PPIB (peptidylprolyl isomerase B (cyclophilin B), PSMB1 (proteosome 26S subunit beta type, 1) PSMD1 (proteosome 26S subunit non-ATPase, 1), RYR3 (ryanodine receptor 3), SOD1 (superoxide dismutase 1), UGDH (UDP-glucose 6-dehydrogenase).

**Figure 9 foods-10-00104-f009:**
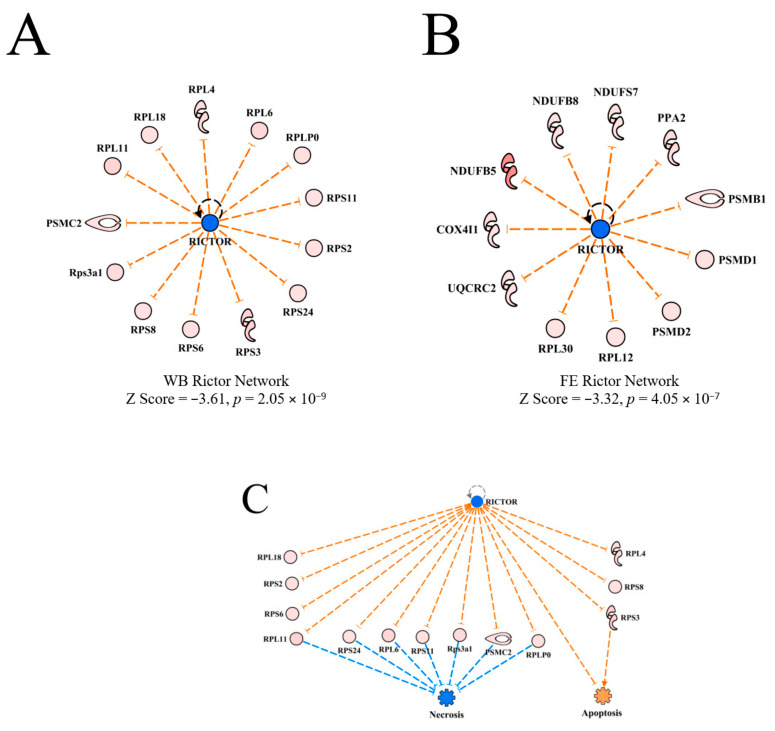
The upstream regulatory network for RICTOR in wooden breast (WB) myopathy (**A**) and for feed efficiency (FE) proteomics (**B**). All downstream proteins were up-regulated in both datasets, but there were no common differentially expressed proteins. The regulatory network for RICTOR linking downstream molecules to the predicted inhibition of necrosis and slight activation of apoptosis in WB myopathy (**C**). For (**A**,**C**), protein abbreviations are defined and differential expression of proteins in WB myopathy are provided in Appendix A. Proteins in (**B**) are as follows: COX4I1 (cytochrome oxidase subunit IV isoform 1), NDUFB5, B8, S7 (NADH dehydrogenase (ubiquinone) Complex I assembly factor 5, factor 8, subunit 7), PPA2 (pyrophosphate (inorganic) 2, PSMB1, D1, D2 (proteosome 26S subunit, non-ATPase beta type-1) PSMD1, PSMD2 (proteosome 26S subunit, non-ATPase, 1 & non-ATPase 2), RPL12, RPL30 (ribosomal protein L12 & L30), UQCRC2 (ubiquinol-cytochrome c-reductase core protein II).

**Figure 10 foods-10-00104-f010:**
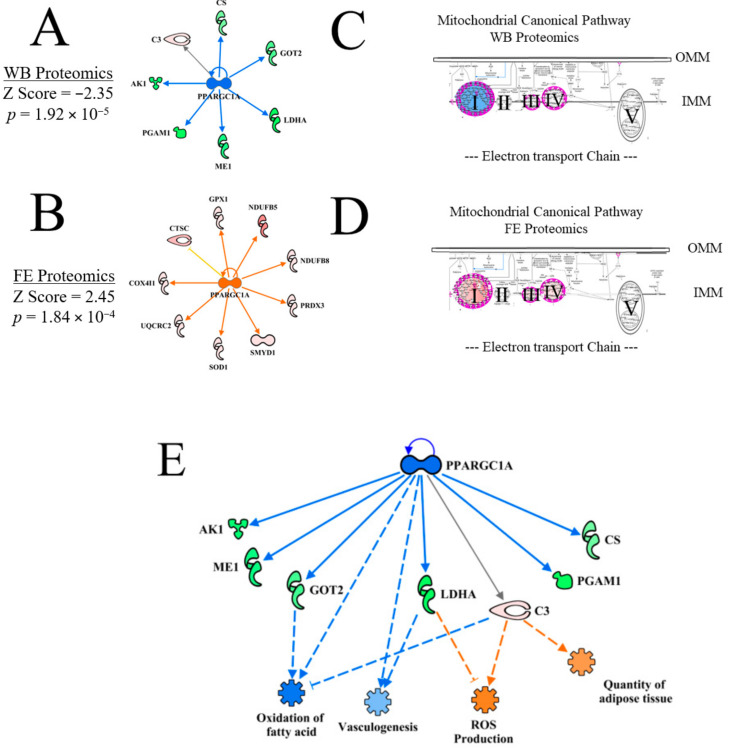
The upstream regulator network for PPARGC1α for WB myopathy vs. control breast muscle in commercial broilers (**A**) and high vs. low FE PedM (**B**) proteomic datasets [11]. The predicted inhibition of PGC1α in WB myopathy was associated with the predicted inhibition of Complex I of the mitochondrial electron transport chain in WB myopathy (**C**), whereas the predicted activation of PGC1α in high FE was associated with predicted activations of Complex I, III and IV of the mitochondrial electron transport chain (**D**). The regulatory network for PPARGC1A showing the downstream target molecules involved in the predictions of inhibition of vasculogenesis and oxidation of fatty acid and activation of reactive oxygen species (ROS) production and quantity of adipose tissue (**E**). Protein abbreviations in (**A**,**E**) are defined and differential expression of proteins are provided in Appendix A. Protein abbreviations for (**D**) are as follows: COX4I1 (cytochrome oxidase subunit IV isoform 1), CTSC (cathepsin C), GPX1 (glutathione peroxidase 1), NDUFB5 (NADH dehydrogenase (ubiquinone) Complex I assembly factor 5), NDUFB8 (NADH dehydrogenase (ubiquinone) beta subcomplex, 8), OMM (Outer Mitochondrial Membrane), IMM (Inner Mitochondrial Membrane).

**Figure 11 foods-10-00104-f011:**
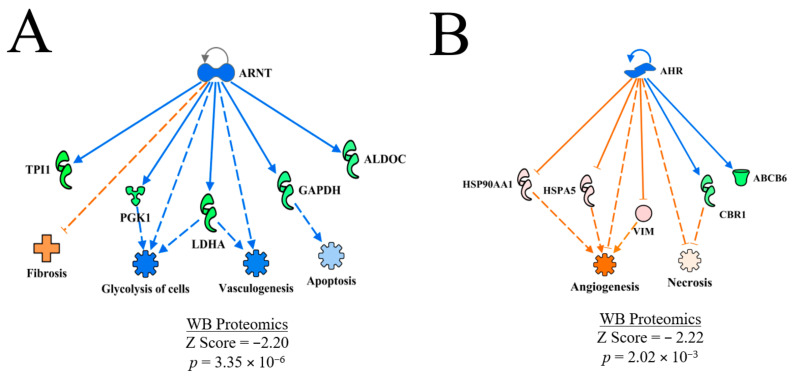
The upstream regulatory networks for ARNT (**A**) and AHR (**B**). The predicted inhibition of ARNT was based on the downstream expression of proteins that are shown and was associated with predicted inhibition of apoptosis, vasculogenesis and glycolysis and prediction of activation of fibrosis. Conversely, the expression of downstream target molecules of involved in predicting inhibition of AHR were associated with predictions of enhanced angiogenesis and necrosis. The discrepancy in the prediction of active necrosis in the AHR and prediction of inhibition of necrosis in the entire dataset shown in Table 2 is based on the smaller subset of expression data for AHR compared to the overall dataset. Protein abbreviations are defined and differential expression of proteins are provided in Appendix A.

**Figure 12 foods-10-00104-f012:**
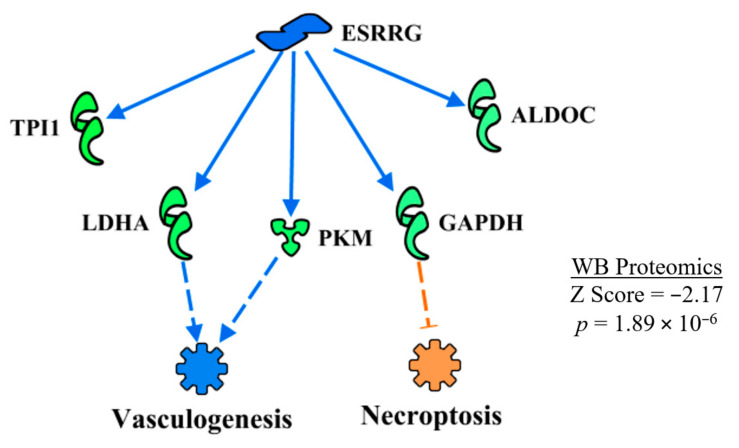
ESRRGamma (ESRRG or ESSRγ) was predicted to be inhibited in WB myopathy based on the expression of downstream molecules shown in this regulatory network. The regulatory network resulted in the prediction of the inhibition of vasculogenesis and activation of necroptosis. As in Figure 9, discrepancies in the predictions of these functions and those in Table 2 are due to a smaller subset of proteins in the ERRG upstream regulatory network, compared to the larger set of proteins listed for functions in Table 2. Protein abbreviations are defined and differential expression of proteins are provided in Appendix A.

**Figure 13 foods-10-00104-f013:**
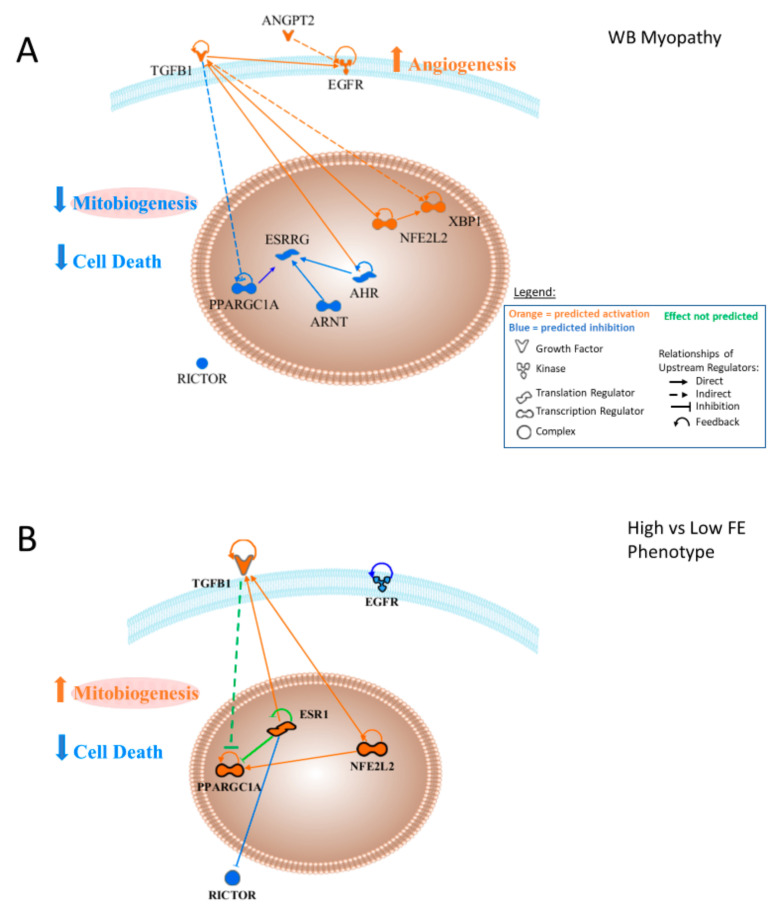
(**A**) Upstream regulator interactions contributing to the overall prediction of increased vasculogenesis and decreased cell death in wooden breast muscle myopathy in commercial broilers. The combination of the predicted activities of these upstream regulators would support the prediction of increased angiogenesis and inhibition of cell death in wooden breast myopathy. (**B**) Arrangement of similar upstream regulators in the high vs. low feed efficiency PedM phenotype dataset revealed a scenario in which angiogenesis was not predicted to be active or inhibited while predictions were made of inhibition cell death, which is similar to that for WB myopathy.

**Table 1 foods-10-00104-t001:** Body weights (BW) and wooden breast (WB) scores in commercial broilers with and without WB, and BW, weight gain (Gain), feed intake (FI) and feed efficiency (FE, gain to feed) in Pedigree Male broilers from which shotgun proteomics was conducted on breast muscle tissue ^1^.

Commercial Broilers ^2^				Pedigree Male Broilers ^3^			
	Normal	Severe WB			High FE	Low FE	
Variable	(*n* = 5)	(*n* = 5)	*p*	Variable	(*n* = 4)	(*n* = 4)	*p*
BW Kg (52 days)	3.2 + 0.2	3.8 + 0.2	0.031	BW (49 days)	3.13 + 0.08	3.18 + 0.03	0.573
WB Score	0.2 + 0.2	1.4 + 0.2	0.003	Gain (kg/7 days)	0.64 + 0.04	0.47 + 0.04	0.022
				FI (kg/7 days)	0.99 + 0.06	1.03 + 0.09	0.689
				FE (G:F)	0.65 + 0.01	0.46 + 0.01	<0.0001

^1^ Values represent the mean ± SE. ^2^ Kuttappan et al. [2]; ^3^ Bottje et al. [13]. Visual scoring values of WB (ranging 0–3) for samples selected for proteomic analysis based on histological assessment indicating normal tissue and severe WB myopathy.

**Table 2 foods-10-00104-t002:** Functions predicted to be activated (orange) or inhibited (blue) in wooden breast (WB) myopathy compared to normal muscle based on differentially expressed (DE) target molecules in commercial broilers.

Functions	Activation *z*-Score	*p*-Value of Overlap	Differentially Expressed Proteins ^1^
Development of vasculature	2.45	1.21 × 10^−3^	ACP1, ANXA2, C3, CALR, CAPNS1, CDH13, CRYAB, CTSB, FGA, FLNA, HSPA5, IGHM, LDHA, PGK1, PKM, PPIA, RHOA, VIM, YWHAZ
Vasculogenesis	2.43	4.46 × 10^−4^	ACP1, ANXA2, C3, CALR, CAPNS1, CDH13, CRYAB, CTSB, FLNA, HSPA5, LDHA, PKM, PPIA, RHOA, VIM, YWHAZ
angiogenesis	2.30	1.67 ×10^−3^	ACP1, ANXA2, C3, CALR, CAPNS1, CDH13, CRYAB, CTSB, FLNA, HSPA5, LDHA, PGK1, PKM, PPIA, RHOA, VIM, YWHAZ
Necrosis	−3.33	2.68 × 10^−11^	ACP1, ANXA2, C3, CALR, CAPNS1, CBR1, CRYAB, CS, CTSB, DDX3X, DYNC1H1, EEF1A1, EIF3L, FDPS, FGA, FLNA, GAPDH, GLO1, HINT1, HK1, HNRNPH1, HSP90AA1, HSP90B1, HSPA5, IGHM,KLHL40, LDHA, P4HB, PARK7, PDCD6IP, PDIA3, PEBP1, PKM, POSTN, PPIA, PRDX6, PSMC2, RACK1, RAN, RHOA, RPL11, RPL6, RPLP0, RPS11, RPS14, RPS24, RPS3, Rps3a1, RPS6, RPS7, TAGLN2, UBE2V2, VIM, YWHAE, YWHAZ
Cell death	−3.05	2.01 × 10^−9^	ACP1, AK1, ALDOC, ANXA2, C3, CALR, CAPNS1, CBR1, CRYAB, CS, CTSB, DDX3X, DES, DYNC1H1, EEF1A1, EIF3L, FDPS, FGA, FLNA, GAPDH, GLO1, HINT1, HK1, HNRNPH1, HSP90AA1, HSP90B1, HSPA2, HSPA5, IGHM, KLHL40, LDHA, P4HB, PARK7, PDCD6IP, PDIA3, PEBP1, PKM, POSTN, PPIA, PRDX6, PSMC2, RAB1A, RACK1, RAN, RHOA, RPL11, RPL6, RPLP0, RPS11, RPS14, RPS24, RPS3, Rps3a1, RPS6, RPS7, TAGLN2, UBE2V2, VIM, YWHAE, YWHAZ
Apoptosis	−1.98	4.61 × 10^−6^	ALDOC, ANXA2, C3, CALR, CAPNS1, CRYAB, CS, CTSB, DDX3X, DES, DYNC1H1, EEF1A1, FLNA, GAPDH, GLO1, HINT1, HK1, HNRNPH1, HSP90AA1, HSP90B1, HSPA2, HSPA5, IGHM, KLHL40, LDHA, P4HB, PARK7, PDCD6IP, PDIA3, PEBP1, PKM, PPIA, PRDX6, RACK1, RHOA, RPLP0, RPS24, RPS3, RPS6, TAGLN2, UBE2V2, VIM, YWHAE, YWHAZ

^1^ Target proteins in plain red type were up-regulated (*p* < 0.05, >1.3-fold difference) in WB myopathy compared to normal tissue. Target molecules in bold green and underlined type were down-regulated (*p* < 0.05, <−1.3-fold difference) in WB myopathy compared to normal tissue.

**Table 3 foods-10-00104-t003:** Upstream regulators predicted to be activated (orange) or inhibited (blue) in wooden breast (WB) myopathy compared to normal muscle based on differentially expressed downstream target molecules in commercial broilers.

Upstream Regulator	Protein Name	Activation *z*-Score	*p*-Value of Overlap	Differentially Expressed Downstream Target Proteins ^1^
ANGPT2	Angiopoeitin 2	3.08	7.41 × 10^−9^	CALR, CRYAB, HSP90AA1, HSPA2, HSPA5, P4HB, PDIA3, PDIA6, POSTN, RPS6
EGFR	Epidermal growth factor	2.62	1.38 × 10^−3^	CRYAB, HNRNPH1, HSP90B1, HSPA5, POSTN, PPIA, VIM
XBP1	X box binding protein 1	2.41	8.79 × 10^−3^	CALR, HSP90B1, HSPA5, PDIA3, PDIA4, PDIA6
TGFB1	Transforming growth factor beta 1	2.21	1.95 × 10^−6^	AK1, ANXA2, CAPNS1, COL7A1, CTSB, DES, EEF1A1, FLNA, FSCN1, GATD3A/GATD3B, HNT1, HNRNPH1, HSP90A1, HAPA5, IARS, IGHM, LDHA, PGK1, PLS3, POSTN, RAB1A, RACK1, RHOA, VIM, YWHAE
NFE2L2	Nuclear factor (erythroid-derived 2)-like 2	2.18	4.3 × 10^−9^	ARF1, CBR1, DSTN, HSP90AA1, HSP90B1, ME1, PDIA3, PDIA4, PDIA6, PGD, RACK1, RAN, RPL18, RPLPO, TPI1
RICTOR	Regulatory-associated protein Independent of mTOR complex 2	−3.61	2.06 × 10^−9^	PSMC2, RPL11, RPL18, RPL4, RPL6, RPLP0, RPS11, RPS2, RPS24, RPS3
PPARGC1A	Peroxisome proliferator activator receptor gamma-coactivator 1 α	−2.35	1.92 × 10^−4^	AK1, C3, CS, GOT2, LDHA, ME1, PGAM1
AHR	Aryl hydrocarbon receptor	−2.22	2.02 × 10^−3^	ABCB6, CBR1, GOT1, HSP90AA1, HSPA5, IGHM, VIM
ARNT	Aryl hydrocarbon receptor nuclear translocator	−2.20	3.35 × 10^−6^	ALDOC, GAPDH, IGHM, LDHA, PGK1, TPI1, VIM
ESRRG (NR3B3)	Estrogen related receptor gamma	−2.17	1.89 × 10^−6^	ALDOC, GAPDH, LDHA, PKM, TPI1

^1^ Target molecules in plain red type were up-regulated (*p* < 0.05, >1.3-fold difference) in WB myopathy compared to normal tissue. Target molecules in bold green and underlined type were down-regulated (*p* < 0.05, <−1.3-fold difference) in WB myopathy compared to normal tissue. ANGPT2: Angiopoeitin 2; EGFR: epidermal growth factor; XBP1: X box binding protein; TGFB1: transforming growth factor beta 1; NFE2L2: nuclear factor (erythroid-derived 2)-like 2.

**Table 4 foods-10-00104-t004:** Upstream regulators that were predicted to be activated (**+**) or inhibited (**--**) in wooden breast myopathy (WB) proteomics (from Kuttappan et al. [2]) with the resulting prediction when the FE proteomics dataset from high and low feed efficiency (FE) Pedigree Broilers males [11] was overlaid on the WB upstream regulator network. The designation ‘no’ indicates that no prediction was made with the FE overlay (projection of FE dataset onto WB myopathy dataset).

Upstream Regulator	WB	FE Overlay	Upstream Regulator	WB	FE Overlay
ANGPT2	**+**	no	RICTOR	**--**	no
EGFR	**+**	no	PPARGC1α	**--**	no
XBP1	**+**	no	AHR	**--**	no
TGFB1	**+**	**+**	ARNT	**--**	no
NFE2L2	**+**	**+**	ESRRG	**--**	no

## Data Availability

No new data were created in this study. Data sharing is not applicable to this article. Support for this research was provided by the USDA-NIFA (#2013-01953) to W.B., B.K. and N.H., USDA-NIFA SAS (#2019 69012-29905) to W.B., S.D., B.K. and N.H. and the Arkansas.

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
