# Peer review of "Upstream Regulator Analysis of Wooden Breast Myopathy Proteomics in Commercial Broilers and Comparison to Feed Efficiency Proteomics in Pedigree Male Broilers"

_foods, 2021, doi:10.3390/foods10010104_

Round 1

Reviewer 1 Report

  1. The comparison between the effects of Normal vs. WB and High-FE vs. Low-FE are not justified biologically. Moreover, such comparisons (if justified) must be conducted using groups of broilers that differ in WB and in FE, but share the same genetic background and reared together. In the submitted manuscript, the two pairs of contrasting groups differ considerably in their genetic backgrounds, and were reared in separate trials. Therefore, any difference (or lack of difference) could be attributed to genetic backgrounds rather than to the contrasting phenotypes.
  2. There were only 5 birds per WB group and 4 birds per FE group. These numbers are barely statistically sufficient for the differences in the contrasting phenotypes, but there are no indications that these small groups represent the proteomic variability of normal vs. WB broilers, and Low-FE vs. High-FE broilers. Moreover, the lack of expressions of proteomic variability, and the wording of M&M, suggest that the breast muscles samples of the 4 or 5 birds in each group were pooled for the proteomic analysis… Was it indeed so?
  3. With only 4 or 5 birds per group, the individual phenotypic values MUST be presented, to clarify obscure means and SE values, such as…
    a. With WB score from 0-1 (normal) to 3 (severe) [line 148], for the 5 Normal birds with mean=0.2 and SE=2, the individual scores must be 0,0,0,0,1 as expected. However, for the 5 WB birds with mean 1.4 and SE=0.2, the individual scores must be 1,1,1,2,2. These scores do not justify the title "Severe WB"!!
    b. The SE of Gain and FI in the Low-FE group are about 10-times large than the comparable SE in the High-FE group. Something MUST be wrong with these data!
  4. The interpretation of the difference between groups within the two pairs must take into consideration the followings:
    a. With almost 20% difference in BW between the WB and Normal groups, proteomic differences between these two groups may reflect confounding effects of WB and BW.
    b. For a valid comparison of FE, the tested birds must have similar BW at the beginning of the FI-test period, to assure similar starting levels of feed consumed for body maintenance.

Author Response

My responses to each of the criticisms or concerns are provided below. Thank you very much for your time and energy in reviewing this manuscript. 

1) Comparisons of WB and FE proteomics datasets are not biologically justified.

It is absolutely true that the birds had different genetic backgrounds and were raised under separate conditions.  In addition, the proteomics analysis were done in different times and different laboratories.   There is nothing that can be done to change these conditions or the genetics of the two different phenotypes. WB myopathy is believed to result from intense selection pressure for growth performance.  But datasets in the public domain can be down loaded and compared to results obtained in any specific lab.  The Ingenuity Pathway Analysis program allows for comparisons of global expression datasets to facilitate interpretation of data or understanding of a given condition in health, disease, cell lines, etc.  We have presented a new approach to looking at WB myopathy by presenting the upstream analysis that was not discussed or presented in the previous paper by Kuttappan et al. (2017).  Contrasting the upstream analysis and regulatory networks with that in the PedM FE dataset presents a different way of looking at this muscle myopathy.   

2) The number of birds in each study. 

This again are the conditions of the two studies and something that can't be changed at this point.  The number of observations in the two papers in which the proteomic data were first reported were not an problem encountered with reviewers in the two journals; Poultry Science and PLoS One.  But beyond that, we were able to look at hundreds of proteins in each animal that can have impacts on understanding of the overall biology of the animal models that can make up for the low number of birds in each study. 

3) a. Designation of WB scoring 

This was not explained clearly - my apology. It appears that this could also have been more clearly presented in the Kuttappan et al. (2017) paper.  After the birds were killed, the breast muscle was examined for visual and physical scoring of WB myopathy.  Tissue samples were then obtained from the bird and a portion flash frozen in liquid nitrogen and another portion obtained for histological analysis. The designation of normal or severe WB myopathy was based on the histological examination.  Of all the samples that were obtained, 5 were chosen for normal histology and 5 others chosen representing severe WB myopathy. After making the designations, the visual and mechanical scores of the two groups were determined.  This procedure is outlined in the paper by Kuttappan et al. (2013). 

To make this more clear in the present study, a third footnote has been added to Table 1 and headings were changed to indicate 'WB" or 'Normal' in the column headings.   

Thank you for pointing this out.  

3) b. SE values in the FE study.

The SE values in the high FE group were in error. This has been corrected on L157 and L158 in Table 1 (highlighted in blue).  Thank you very much for catching this error. 

4) a. Difference in body weight in commercial broilers. 

This actually highlights a major aspect of WB myopathy.  It is mainly fastest growing, highest performing birds that exhibit myopathy in a flock.  There is concern that selection for growth performance (i.e. greater growth and feed effiicency) is a main reason for this myopathy that has appeared in the last decade.  And this is one of the reasons that this examination of proteomics from WB in commercial broilers and in PedM broilers was conducted - to see if similarities and differences exist in the proteomic millieu of these two animal models.  It is difficult to find birds with and without myopathy and the same age and BW.  Invariably, those with WB myopathy at a given age weigh more than those with no myopathy. 

4) b. Similar body weight at the beginning of FE phenotyping period. 

The birds that were investigated in the FE study were obtained from one of the largest commercial broiler breeding companies in the world.  This company has been conducting FE phenotyping 3 times per week for over 20 years between 6 and 7 weeks of age. In this, and other studies that we conducted with this pedigree male broiler line, the high FE birds gained more weight during the week of phenotyping, but did so consuming the same amount of feed as the Low FE phenotype.  Invariably, these high FE birds would start the week with lower BW but would end the week with the same BW as the low FE group; they gained more BW on the same amount of feed.   So - starting at the same BW was not possible in this animal model. 

Reviewer 2 Report

The Authors have investigated an interesting topic, and the theme has been properly described. I would like to congratulate Authors for the good-quality of their article, the literature reported used to write the paper, and for the clear and appropriate structure. The manuscript is well written, presented and discussed, and understandable to a specialist readership.

In general, the organization and the structure of the article are satisfactory and in agreement with the journal instructions for authors. The subject is adequate with the overall journal scope. The work shows a conscientious study in which a very exhaustive discussion of the literature available has been carried out. The introduction provides sufficient background, and the other sections include results clearly presented and analyzed exhaustively.

As specific comment, I suggest to check the references list, there are some typos in citing papers and/or journal names.

Author Response

Thank you for the review.  I apologize for the typos in the references - hopefully all have been caught and corrected. 

Reviewer 3 Report

Dear authors, the present study provides new and important data about wooden breast myopathy and feed efficiency related proteomics in commercial and pedigree broilers respectively. More specifically the analysis for AHR, ARNT and HSP90, their interactions and the metabolic pathways which are involved, could give a clear view about their importance in broilers physiology. The present work could be considered also as a review due to the analytically presentation of the investigated factors.

However, there are some important issues which must be answered according to the zootechnical methods which were used.

Firstly in L. 79-80 it is mentioned that n=24 broilers were randomly selected at 52d of a previous study. I recommend you to mention the experimental treatments and the basal diets which were used in the study from which broilers were randomly taken.

Moreover, I would like to ask why broilers were taken at 52d of the experiment although in the majority of broilers studies the experimental periods last from 35d to 42d maximum. Moreover, in L. 90 you believe that n=4 broilers per group are enough in order to investigate feed efficiency proteomics?

In addition, in L. 165 breast muscle samples were taken at 52d for WB study and between 56 and 63d for FE study. Why for the same reasons as in L. 80 samples were not taken earlier (i.e. at 42d which is commonly used in practice)?

Finally in L. 573, it is mentioned that in the present study all of the predictions presented are just predictions. This work could also be a very good review, instead of an analytical study.

Conclusively, the present study is recommended to be minor revised.

Author Response

Thankyou for your comments.  I will address each one and indicate where changes have been made to the manuscript.

1) L79-80

Birds in both studies received diets that met or exceeded the nutrient requirements for broilers containing approximately 20% protein and 3200 kcal/kg.   This has been added to the manuscript on L80-82 and on lines 92-93 (in bold type) 

2) Why were birds sampled on day 52 in the wooden breast proteomic study (and day 49 in PedM feed efficiency study)?

In the commercial broiler industry, many integrators are growing birds for 8 to 10 weeks for a large bird bird (8-10 lbs).  The incidence of Wooden breast myopathy increases in older and heavier birds.  This is the main reason birds were sampled at this age.  

Birds in the FE study were sampled after 7 weeks (day 49+) since the pedigree male broilers are phenotyped for feed efficiency between 6 to 7 weeks. 

3) Do we think the n of 4 or 5 is sufficient for these studies?

The short answer is yes.  Differential expression of proteins in the study was set at P < 0.05 and 1.3 fold different between groups.   In the present study the predictions of activation or inhibition of upstream regulators had P values  that were less than 0.000.1 in most cases.  This level of significance is possible because of the large amount of proteins that were detected and their interactions in various pathways or cellular networks.

I would also like to indicate that the number of birds from these studies were sufficient for the two journals in which the research was first reported: Poultry Science and PLoS One.  The datasets are both in the public domain and available to anyone that would to have interest in accessing these datasets. 

4) Should this manuscript be a review paper rather than an analytical study?

I think that even though these are technically predictions made by the software, it is based on analytical tools and should be considered more of a mathematical or statistical analysis and therefore would be analytical rather than a review.  I hope that makes sense.  I can elaborate further if necessary.  If you feel this should be added to the discussion at the end, I would be happy to do so. 

Round 2

Reviewer 1 Report

Based on my expertise in statistics and in the aspects of wooden breast (WB) and feed efficiency (FE) in broilers, I cannot approve Table 1 as it is now, for the following reasons:

1. The wording of the * footnote ["Mean values in a row and broiler type are higher (P<0.05)"] is wrong. What they mean by the word "higher"?... It must be re-written.

2. I cannot accept the authors' claim that the High FE and Low FE groups differ significantly in their Gain. The difference between the means of these two groups ia 0.17 (0.64-0.47), whereas the standard errors of these means (0.40 and 0.38) are much larger, and therefore it is IMPOSSIBLE that a difference of 0.17 was significant.

3. There were only 5 replicates (individual broilers) in the two WB groups, and only 4 individual broilers in the FE groups. The authors must show these numbers in Table 1.

4. I realize that the authors cannot change these small numbers, but in their manuscript they MUST refer to the limited reliability of their findings due to low number of replicates.

Author Response

Response to Reviewer 1

I want to thank the reviewer for being persistent with the comments that were made.  I am not sure where the mistake was made, but the version of the revised manuscript that was loaded was not the final version.  I agree with the reviewer, that based on the standard error values that were provided, the means would not be significantly different.   I have made changes that should answer these concerns that includes clear indication of the number of observations in each group.

With regard to concern about the numbers of samples that were analyzed in both studies being representative of the phenotypes or having limited reliability, I have tried to address these concerns in two ways.

1) By inserting P values in the table

2) By adding text (indicated in blue bold text) on lines 171 to 175.